

# RC4USCoast: A river chemistry dataset for regional ocean model applications in the U.S. East, Gulf of Mexico, and West Coasts

Fabian A. Gomez[1,2], Sang-Ki Lee[2], Charles A. Stock[3], Andrew C. Ross[3], Laure Resplandy[4], Samantha A. Siedlecki[5], Filippos Tagklis[2,6], Joseph E. Salisbury[7]

[1]Northern Gulf Institute, Mississippi State University, Starkville, Mississippi, USA.

[2]NOAA Atlantic Oceanographic and Meteorological Laboratory, Miami, Florida, USA.

[3]NOAA Geophysical Fluid Dynamics Laboratory, Princeton, New Jersey, USA

[4]Department of Geosciences, High Meadows Environmental Institute, Princeton University, Princeton, New Jersey, USA

[5]Department of Marine Sciences, University of Connecticut, Groton, Connecticut, USA

[6]Cooperative Institute for Marine and Atmospheric Studies, University of Miami, Miami, Florida, USA.

[7]Ocean Process Analysis Laboratory, University of New Hampshire, Durham, New Hampshire, USA

*Correspondence to*: Fabian A. Gomez (fabian.gomez@noaa.gov)

**Abstract.** A historical dataset of river chemistry and discharge is presented for 140 monitoring sites along the United States
East Coast, the Gulf of Mexico, and the West Coast from 1950 to 2020. The dataset, referred to here as River Chemistry for
the U.S. Coast (RC4USCoast), is mostly derived from the Water Quality Database of the U.S. Geological Survey (USGS), but
also includes river discharge from the USGS's Surface-Water Monthly Statistics for the Nation and the U.S. Army Corps of
Engineers. RC4USCoast provides monthly time series as well as long-term averaged monthly climatological patterns for
twenty variables including alkalinity and dissolved inorganic carbon concentration. It is mainly intended as a data product for
regional ocean biogeochemical models and carbon chemistry studies in the U.S. coastal regions. Here we present the method
to derive RC4USCoast and briefly describe the river's carbonate chemistry patterns.

## 1 Introduction

Riverine fluxes of water, nutrients, alkalinity, and carbon exert a significant impact on the coastal ocean margins, modulating
patterns in primary production, dissolved oxygen, calcium carbonate saturation, bottom acidification, and air-sea carbon fluxes
(e.g., Rabouille et al., 2008; Cai et al., 2013; Siedlecki et al. 2017; Moore-Maley et al., 2018; Xie et al., 2020; Liu et al., 2021).
During the last decade or so, there has been an increasing interest in better understanding and quantifying the influence of river
inputs on the coastal ecosystems of the United States. This is reflected in a growing number of ocean biogeochemical (BGC)
modeling studies addressing river-induced ocean patterns (e.g., Fennel et al, 2011; 2013; Laurent et al., 2017; Siedlecki et al.



2017; 2021; Hood et al., 2021; Gomez et al., 2021). Ocean BGC models need realistic inputs of river-water properties to
properly simulate coastal ecosystem responses to river runoff, but the availability of these inputs is usually limited (e.g.,
Kearney et al., 2021). A few existing data products contain estimates of riverine carbon and/or nutrients based on empirical or
dynamic river export models (e.g., Mayorga et al., 2010; Li et al., 2017; 2019; Lacroix et al., 2021; Regnier et al., 2022). These
products were mainly developed for global budget analysis, and consequently they often lack sufficient spatial resolution to
allow the study of ecosystem dynamics at a regional scale or have significant regional biases. Motivated by the necessity of
high-resolution river chemistry data for regional ocean BGC models, here we present the River Chemistry for the U.S. Coast
(RC4USCoast) database, a compilation of historical river chemistry and discharge records derived from the U.S. Geological
Survey (USGS).

**2 Dataset**

The RC4USCoast database contains historical river chemistry records from 140 USGS monitoring stations retrieved from the
Water Quality Database of the National Water Information System (Alexander et al., 1998;
https://nwis.waterdata.usgs.gov/usa/nwis/qwdata). We use a set of stations similar to those used in Stets and Striegl (2012),
who selected stations based on the availability of water quality records and proximity to river mouths. These monitoring
stations correspond to 52 rivers in the US East Coast, 53 rivers in the Gulf of Mexico, and 35 rivers in the US West Coast (Fig.
1; Table S1 in the Supplement). It is worth noting that Stets and Striegl (2012) reported average inorganic and organic carbon
flux (g C yr$^{-1}$) and yield (g C m$^{-2}$ yr$^{-1}$) for the selected USGS stations, but they did not provide a dataset with the riverine
concentration of carbon. Therefore, RC4USCoast advances providing integrated information on river DIC and alkalinity
concentration (Sect. 2.1) and, where available, additional inorganic and organic nutrients relevant for coastal water quality
(Sect. 2.2) for those stations.

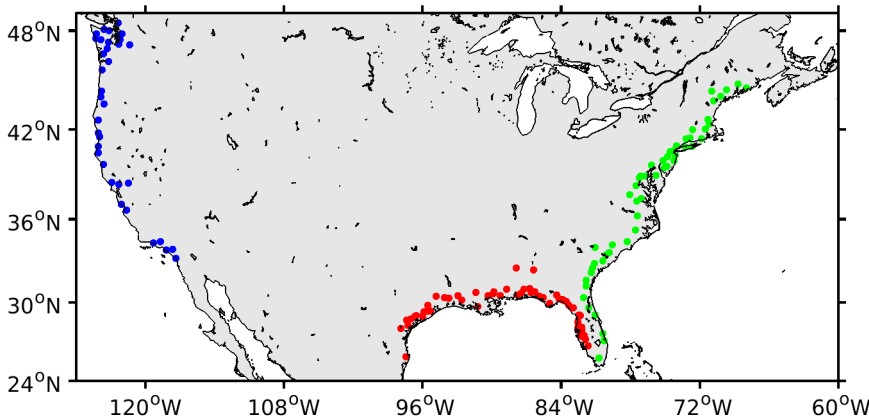


**Figure 1.** USGS stations used to derive river chemistry patterns. Green, red, and blue dots correspond to river discharging to
the East, Gulf of Mexico, and West Coast, respectively.



## 2.1 Carbon chemistry

RC4USCoast includes a river carbon chemistry dataset with monthly series and climatological data for alkalinity, pH field, pH
laboratory, DIC, and dissolved organic carbon (DOC) (Table 1). To this effect, we processed more than 61,000 records of
calcium carbonate ($CaCO_3$) and bicarbonate ($HCO_3$), 56,000 pH field and laboratory records, and 8,000 DOC records. Due to
the substantially smaller number of DIC measurements (~1,800) compared to those of alkalinity and pH, we derived DIC from
alkalinity, pH, and water temperature using the CO2SYS program for CO2 System Calculations (van Heuven et al., 2011).
Following Stets and Striegl (2012), we assumed that (*i*) particulate inorganic carbon is small; thus, filtered and unfiltered
measurements of alkalinity are nearly the same, and (*ii*) inorganic carbon represents the major fraction of river alkalinity. A
comparison between filtered and unfiltered measurements of alkalinity does not show significant differences (Fig. 2a); thus,
biases associated with the first assumption are negligible. The second assumption implies that DIC estimates do not account
for non-carbonate alkalinity, which may lead to DIC overestimation. This is especially true in low alkalinity rivers with high
concentration of organic matter, as the latter contains anionic functional groups that can contribute to alkalinity (Hunt et al.,
2011). Stets and Striegl (2012) discussed this issue further and showed that ignoring the non-carbonate alkalinity usually led
to an overestimation of DIC <10%. Consistently, a comparison between measured DIC and the calculated DIC reveals a good
agreement, with no evident bias in the residuals of the least square model (Fig. 2b).

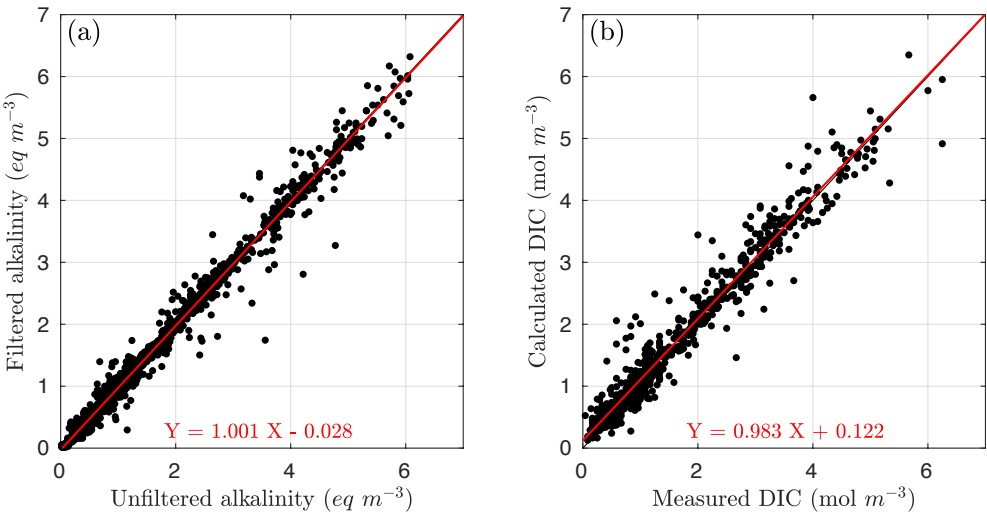


**Figure 2.** Data comparison: (a) Filtered vs. unfiltered alkalinity; (b) measured vs. calculated DIC. Calculated DIC was derived
from alkalinity, pH, and temperature measurements.



**Table 1.** Carbon system variables in the RC4USCoast dataset.

| Variable | Units | USGS parameter code | Description | Original USGS units | Water chemistry measurements |
|---|---|---|---|---|---|
| Alkalinity | meq m$^{-3}$ | 00410 | Acid neutralizing capacity, unfiltered, fixed endpoint titration, field | mg CaCO3 liter$^{-1}$ | 20,427 |
| | | 00419 | Acid neutralizing capacity, unfiltered, inflection-point titration, field | mg CaCO3 liter$^{-1}$ | 378 |
| | | 29801 | Alkalinity, filtered, fixed endpoint titration, laboratory | mg CaCO3 liter$^{-1}$ | 2,839 |
| | | 39036 | Alkalinity, filtered, fixed endpoint titration, field | mg CaCO3 liter$^{-1}$ | 587 |
| | | 39086 | Alkalinity, filtered, inflection-point titration, field | mg CaCO3 liter$^{-1}$ | 6,428 |
| | | 90410 | Acid neutralizing capacity, unfiltered, fixed endpoint titration, laboratory | mg CaCO3 liter$^{-1}$ | 8,581 |
| | | 00440 | Bicarbonate, unfiltered, fixed endpoint titration, field | mg HCO3 liter$^{-1}$ | 16,121 |
| | | 00453 | Bicarbonate, filtered, fixed endpoint titration, field | mg HCO3 liter$^{-1}$ | 6,330 |
| pH field | standard units | 00400 | pH, unfiltered, field | standard units | 43,432 |
| pH lab | standard units | 00403 | pH, unfiltered, laboratory | standard units | 13,354 |
| DOC | mmol C m$^{-3}$ | 00681 | Organic carbon, filtered | mg C liter$^{-1}$ | 8,114 |
| DIC | mmol C m$^{-3}$ | | DIC derived from alkalinity, pH, and temperature | | |


## 2.2 Other chemistry variables

The RC4USCoast database also contains a set of variables that describe the runoff of nitrogen, phosphorus, and silica (Table 2), including monthly time series of nitrate ($NO_3$), nitrate plus nitrite ($NO_3$ plus $NO_2$), ammonia ($NH_4$), organic nitrogen plus ammonia (orgN), dissolved organic nitrogen (DON), total nitrogen (TN), phosphate ($PO_4$), total phosphorus (TP), and silicon

dioxide ($SiO_2$). For orgN, TN, and TP, we generated two independent datasets for unfiltered and filtered water samples (the former containing both dissolved and particulate material, and the latter only dissolved material). For $NO_3$, $NO_3$ plus $NO_2$, $NH_4$, and $PO_4$ we considered the USGS parameters for filtered water samples. In addition to these inorganic and organic nutrients, we also included dissolved oxygen (DO) and water temperature in the database.



**Table 2.** Additional variables in the RC4USCoast dataset.

| Variable | Units | USGS parameter code | Description | Original USGS units | Water chemistry measurements |
|---|---|---|---|---|---|
| NO$_3$ | mmol N m$^{-3}$ | 00618 | Nitrate, filtered | mg N liter$^{-1}$ | 23,692 |
| | | 71851 | Nitrate, filtered | mg NO3 liter$^{-1}$ | 23,593 |
| NO$_3$ plus NO$_2$ | mmol N m$^{-3}$ | 00631 | Nitrate plus nitrite, filtered | mg N liter-1 | 19,939 |
| NH$_4$ | mmol N m$^{-3}$ | 71846 | Ammonia (NH3 + NH4+), filtered | mg NH4 liter$^{-1}$ | 20,091 |
| | | 00608 | Ammonia (NH3 + NH4+), filtered | mg N liter$^{-1}$ | 19,836 |
| Organic nitrogen unfiltered | mmol N m$^{-3}$ | 00625 | Organic nitrogen plus ammonia, unfiltered | mg N liter$^{-1}$ | 21,139 |
| Organic nitrogen filtered | mmol N m$^{-3}$ | 00623 | Organic nitrogen plus ammonia, filtered | mg N liter$^{-1}$ | 10,932 |
| DON | mmol N m$^{-3}$ | 00607 | Dissolved organic nitrogen, filtered | mg N liter$^{-1}$ | 10,250 |
| TN unfiltered | mmol N m$^{-3}$ | 00600 | Total nitrogen [inorganic + organic nitrogen], unfiltered | mg N liter$^{-1}$ | 23,161 |
| TN filtered | mmol N m$^{-3}$ | 00602 | Total nitrogen [inorganic + organic nitrogen], filtered | mg N liter$^{-1}$ | 10,864 |
| PO$_4$ | mmol P m$^{-3}$ | 00660 | Orthophosphate, filtered | mg PO4 liter$^{-1}$ | 20,280 |
| | | 00671 | Orthophosphate, filtered | mg P liter$^{-1}$ | 18,733 |
| TP unfiltered | mmol P m$^{-3}$ | 00665 | Total phosphorous [organic + inorganic phosphorous], unfiltered | mg P liter$^{-1}$ | 26,608 |
| TP filtered | mmol P m$^{-3}$ | 00666 | Total phosphorous [organic + inorganic phosphorous], filtered | mg P liter$^{-1}$ | 19,239 |
| Silica | mmol Si m$^{-3}$ | 00955 | Silica, filtered | mg SiO2 liter$^{-1}$ | 31,940 |
| Dissolved oxygen | mmol O$^2$ m$^{-3}$ | 00300 | Dissolved oxygen, water, unfiltered | mg O2 liter$^{-1}$ | 34,379 |
| Temperature | °C | 00010 | Water temperature | °C | 50,442 |
| Discharge | m$^3$ s$^{-1}$ | 00060 | Mean discharge[a] | ft$^3$ s$^{-1}$ | |
| | m$^3$ s$^{-1}$ | 00061 | Instantaneous discharge[b] | ft$^3$ s$^{-1}$ | |

(a) Averaged discharge from the USGS Surface-Water Monthly Statistics was used for all rivers excepting the Mississippi-Atchafalaya (U.S. Army Corps of Engineers) and those listed in b

(b) Instantaneous discharge from the USGS Water Quality Database was used for the Charles, James, Weeki Washee, and Rio Grande
rivers.



### 2.3 River discharge

To provide a longer set of river discharge records than those available in the USGS Water Quality Database, we used monthly average data from the USGS Surface-Water Monthly Statistics for the Nation database (https://waterdata.usgs.gov/nwis/monthly). Similarly, for the Mississippi and Atchafalaya rivers, we used records from the

U.S. Army Corps of Engineers (USACE). Specifically, we used the Mississippi discharge at the USACE's station 01100 (Tarbert Landing), and the Atchafalaya discharge at station 03045 (Simmesport). Those records were obtained from the discharge dataset in the Gulf of Mexico Coastal Ocean Observing System (GCOOS, https://geo.gcoos.org/river_discharge/). For a few rivers (Charles, James, Weeki Washee, and Rio Grande) where monthly discharge was not available in the USGS Surface-Monthly Statistics database or the USACE records, we used discharge from the USGS Water Quality Database.

### 2.4 Database generation

Information for the selected river stations includes the RC4USCoast river ID, the original USGS site ID, the USGS site's longitude and latitude, and an approximate longitude and latitude for the river mouth (Fig. S1). A few rivers flow to other larger rivers, as described in Table S1. The assigned mouth location in those cases corresponds to the mouth of the major stream discharging to the ocean. For example, the dataset contains the Alabama and Tombigbee rivers, which converge to the

Mobile River, so the associated river mouth for those two rivers is the Mobile mouth (30.7°N and 88.0°W).

To the extent it was possible given data availability, we calculated monthly times series for all variables and all river sites over the period 1950–2020. Temporal data gaps were kept unfilled. In the Water Quality Database, river properties are characterized by a set of parameters, each associated with a specific measurement type. As indicated in Tables 1 and 2, we used eight parameters to derive alkalinity, two parameters to derive $NO_3$, $NH_4$ and $PO_4$, and one parameter for the remaining variables:

pH field, pH laboratory, DOC, $NO_3$ plus $NO_2$, $SiO_2$, DO, temperature, and the filtered and unfiltered concentration of orgN, TN, and TP. Conversion factors were applied to present alkalinity in milliequivalent $m^{-3}$ (meq $m^{-3}$), the carbon-based variables (DIC, DOC) in mmol C $m^{-3}$, the nitrogen-based variables ($NO_3$, $NO_3$ plus $NO_2$, $NH_4$, orgN, TN) in mmol of N $m^{-3}$, the phosphorous-based variables ($PO_4$ and TP) in mmol of P $m^{-3}$, silica in mmol of $SiO_2$ $m^{-3}$, and dissolved oxygen in mmol of $O_2$ $m^{-3}$. To ensure data quality, outliers, defined here as river chemistry values above and below 3.5 standard deviations from the

median were removed. Maximum alkalinity (DIC) values were limited to 8,000 meq (mmol) $m^{-3}$. pH records below 3.5 or above 10 units were discarded. Additionally, an upper threshold of 3.5 was used for the DIC to alkalinity ratio (DIC:Alk ratio), based on values reported by Moore-Maley et al. (2018). DIC records linked to DIC:Alk ratios greater than 3.5 were also removed.



**Figure 3.** Number of monthly records (NMR) in the dataset time series (1950-2020). The colorbar range may vary between panels. Variable description is in Tables 1 and 2.



**Figure 3** (continued).


Except for river discharge, which had average temporal coverage of 85%, the USGS time series had significant data gaps because the parameter's monitoring had a limited number of years, and/or the parameter's measurements were not performed at a regular frequency. Figure 3 displays the number of records (data density) in the monthly time series for each site, indicating large differences among rivers and variables. Monitoring stations with the most complete chemistry records were linked to rivers flowing to the Mid Atlantic Bight, the Mississippi and Atchafalaya, and a limited number of major rivers on the West Coast and Texas coast. The greatest data density was for pH field, water temperature, and alkalinity, with a median of 162, 164, and 139 records (over the 140 sites), respectively, whereas the lowest data density was for DON and DOC, with a median of 21 and 14 records, respectively.

To complement the time series and provide a ready to use dataset for ocean biogeochemical model applications with no data gaps, we generated monthly climatologies using all data during 1950–2020. We also generated climatologies for the 1950–1989 and 1990–2020 periods, as a way to represent temporal variation in the climatological pattern. We considered those multidecadal periods, as the temporal coverage in the river chemistry dataset did not resolve well decadal variability for all sites. To ensure a minimum number of observations to derive the monthly climatologies, for each variable and station we calculated the number of records per calendar month. If the median value of this record count (over the 12 months) was less than five, or any month had no data, then the monthly climatology was substituted by the long-term annual average.

A brief description of the carbon system variables in the RC4USCoast database is provided in the following section. Mean patterns for other variables are shown in the Supplement (Figs. S2).

## 3 Main carbon system patterns

The site-averaged alkalinity concentration ranges from 40 meq m$^{-3}$ (Black water) to 5,605 meq m$^{-3}$ (Santa Clara). The frequency distribution for this variable displays a positive skewness with a median of 662 meq m$^{-3}$ and 42% of the values lower than 500 meq m$^{-3}$ (Fig. 4a). The largest fraction of low alkalinity (<500 meq m$^{-3}$) rivers is on the East Coast, especially for rivers flowing to the Gulf of Maine and South Atlantic Bight (Fig. 5a). On the other hand, the largest fraction of high alkalinity (>2,000 meq m$^{-3}$) rivers is in the Gulf of Mexico (Fig. 4a), mainly clustered over the Texas and West Florida coasts (Fig. 5a). Along the West Coast, there is a clear meridional gradient in river alkalinity, with the highest values in Southern California and the lowest in Oregon and Washington (Fig. 5a).





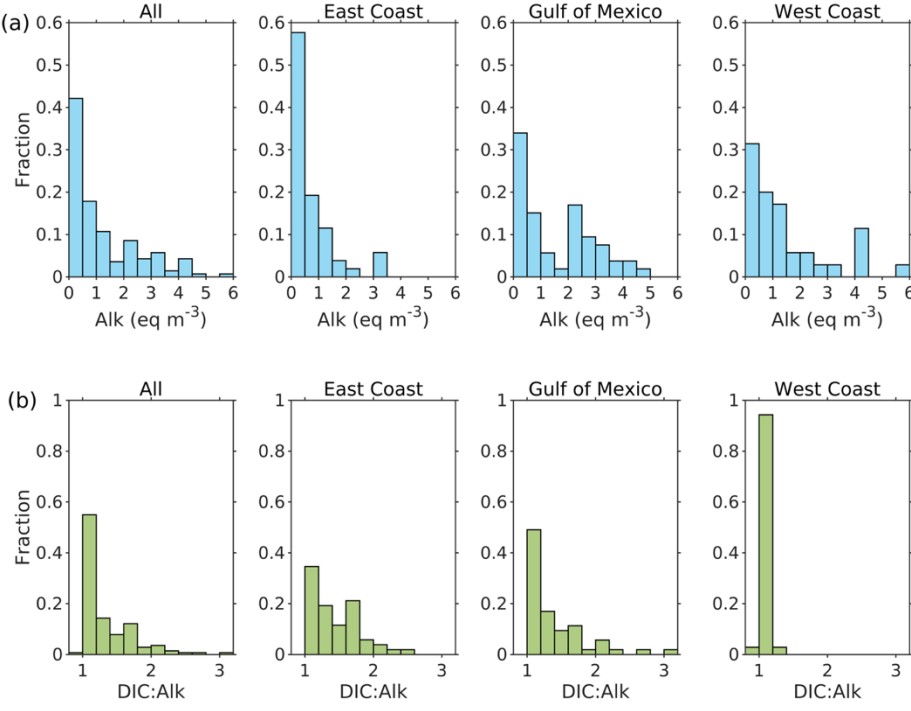

**Figure 4.** Frequency histogram derived from the long-term site-averaged (a) alkalinity and (b) DIC to alkalinity (DIC:Alk) ratio for all rivers (All), and river discharging at the East Coast, Gulf of Mexico, and West Coast.
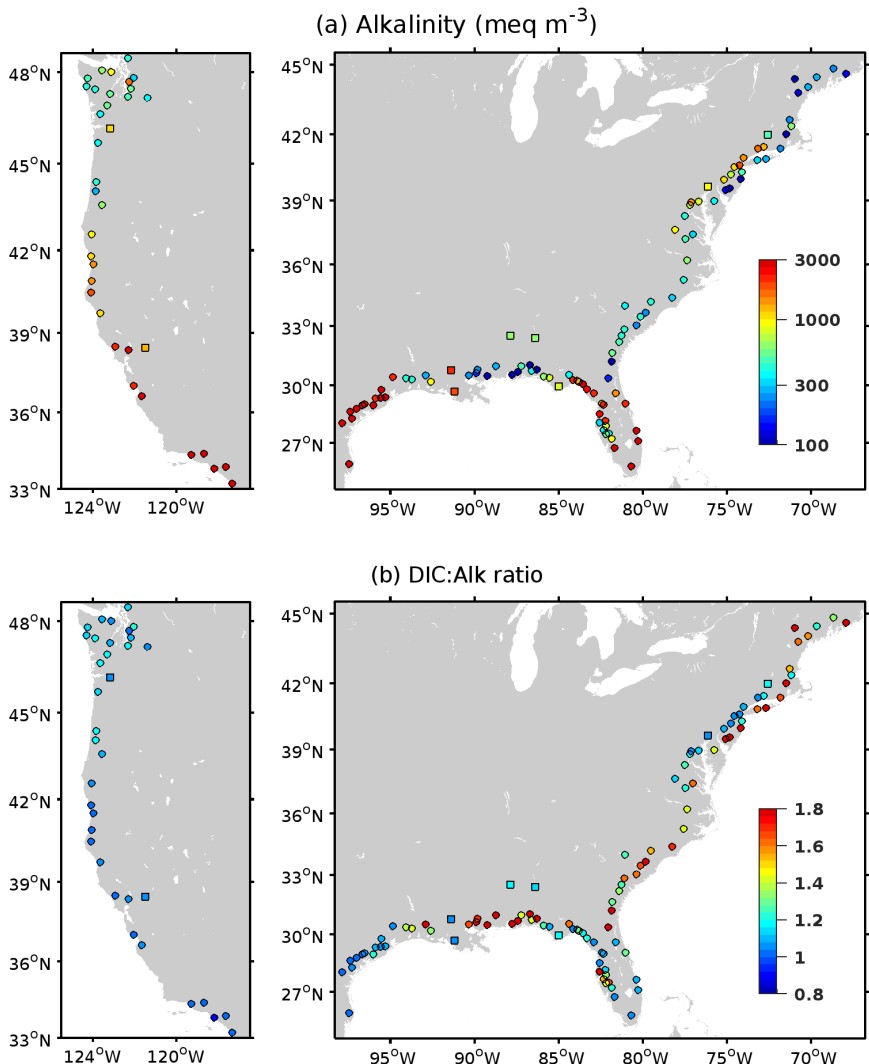

155

**Figure 5.** Long-term mean (colored dots and squares) of the river (a) alkalinity and (b) DIC to alkalinity ratio. Squares (dots) represent river stations with a mean discharge greater (smaller) than 500 m³ s⁻¹. Colorbar in (a) is in logscale.

160

The average river DIC concentration shows a very similar spatial pattern to the average river alkalinity, as both variables are highly correlated ($r = 0.99$; Fig. S3 in the Supplement). However, DIC tends to be greater than alkalinity, which is reflected in an average DIC:Alk ratio of 1.33 over the 140 stations. Like alkalinity, the frequency distribution of the site averaged DIC:Alk ratios has a positively skewed distribution (Fig. 4b), with a median of 1.17, and minimum and maximum values of 0.92 (Los Angeles) and 3.08 (Shoal), respectively. Rivers with the lowest DIC:Alk ratios are in the West Coast, where DIC is on average 8% greater than alkalinity (Fig. 5b). Large DIC:Alk ratios are mainly associated with low alkalinity rivers, and the opposite is true for high alkalinity rivers. Indeed, the relationship between these two variables has a clear linear pattern for alkalinities below ~500 meq m$^{-3}$, where the mean DIC:Alk ratio decreases 0.247 units per every 100 meq alkalinity increase (Fig. 6a). Moreover, we found that the standard deviation of the DIC:Alk ratio (SD$_{DIC:Alk}$) is inversely linked to the mean alkalinity (Fig. 6b). Most rivers with a mean alkalinity below 200 meq m$^{-3}$ have a SD$_{DIC:Alk}$ greater than 0.4, whereas most rivers with a mean alkalinity above 1,000 meq m$^{-3}$ have a SD$_{DIC:Alk}$ lower than 0.2.

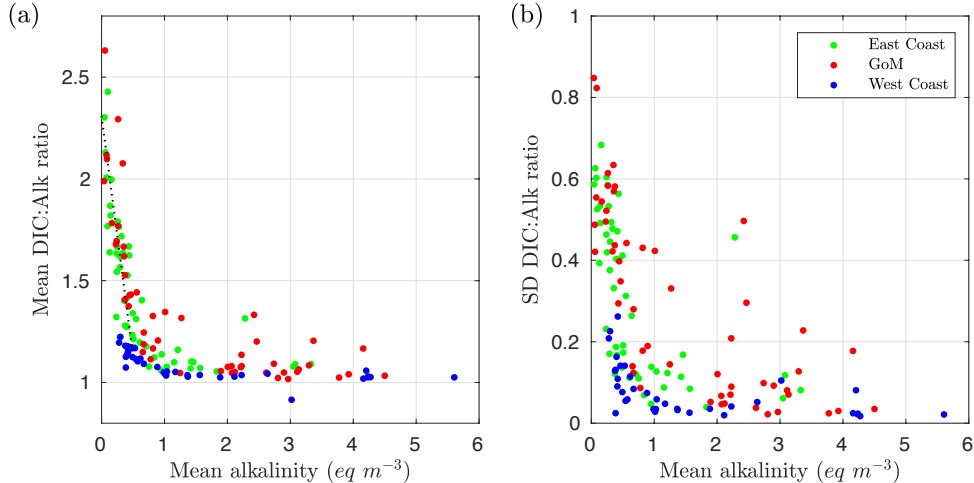

**Figure 6.** Between river variability in the DIC to alkalinity (DIC:Alk) ratio as a function of alkalinity: (a) mean DIC:Alk ratio vs. mean alkalinity; b) standard deviation of the DIC:Alk ratio vs. mean alkalinity. Each dot represents one of 140 rivers in the dataset. Green, red, and blue dots depict the rivers flowing to the East, Gulf of Mexico, and West Coasts, respectively.



To further investigate the river carbon chemistry variability, we examined monthly records for the stations with the largest
data density. Those stations are associated with six rivers in the East Coast (Connecticut, Delaware, Schuylkill, Choptank,
Susquehanna, and Neuse), two rivers in the Gulf of Mexico (Mississippi and Atchafalaya) and four rivers in the West Coast
(Santa Ana, Sacramento, Eel, and Klamath). A strong positive relationship between the monthly alkalinity and DIC records is
evident for all of the 12 stations (Fig. 7a). The coefficients of determination ($R^2$) for the linear regression of DIC against
alkalinity average to 0.91, ranging from 0.57 (Neuse) to 0.99 (Eel) (Table S2 in the Supplement). Like the patterns in Figure
6, the monthly records show an inverse relationship between the DIC:Alk ratio and alkalinity (i.e., an increased variability in
the DIC: Alk ratio at low alkalinity values, and vice versa at high alkalinity values) (Fig.7b). The Choptank and Neuse rivers,
in the lower end of the alkalinity concentration, show the largest dispersion in the DIC:Alk ratio, with values ranging from ~1
to higher than 2.5. In contrast, the high alkalinity Santa Ana River displays a much smaller variability, with the maximum
DIC:Alk ratio around 1.1. Seasonal patterns for alkalinity (and DIC) tend to show enhanced values during summer and fall,
and minimum values during winter and spring (Fig. 7c), concurrent with low and high discharge periods, respectively. This
pattern is consistent with multiple studies conducted in specific river basins suggesting dilution of carbon chemistry variables
during high discharge conditions (e.g., Cai, 2003; Guo et al., 2008; Joesoef et al., 2017). Indeed, a linear relationship between
the logarithm of discharge (logDisc) and alkalinity is evident for each of the 12 stations (Fig. 7d). The adjusted linear regression
models for these rivers are all significant, with linear regression coefficients ranging from 0.34 (Sacramento) to 0.69 (Eel)
(Table S2).

The inverse relationship between logDisc and alkalinity in Figure 7d can be extended to other rivers in the database. Figure 8
shows the regression coefficient (slope) and $R^2$ for the stations where the regression was significant, explained at least 20% of
the alkalinity variance, and include at least 30 observations (77 out of 140 rivers). The sensitivity of river alkalinity to changes
in discharge, reflected in the magnitude of the regression coefficient, is greater in the high alkalinity rivers flowing to the Gulf
of Mexico, Southern California, and East Florida coasts, and smaller in the low alkalinity rivers flowing to the Northwest and
East Coasts (Fig. 8a). This determines a significant negative correlation between the regression coefficient and the site-
averaged alkalinity ($r$ = -0.83). The $R^2$ coefficient pattern shows an important spatial variability (Fig. 8b), which is not linked
to river alkalinity or discharge. The largest $R^2$ values (>0.5) characterize rivers flowing to the Northwest Coast, Florida
Panhandle, and South Atlantic Bight. Similar patterns were found for the relationship between logDisc and DIC (not shown).

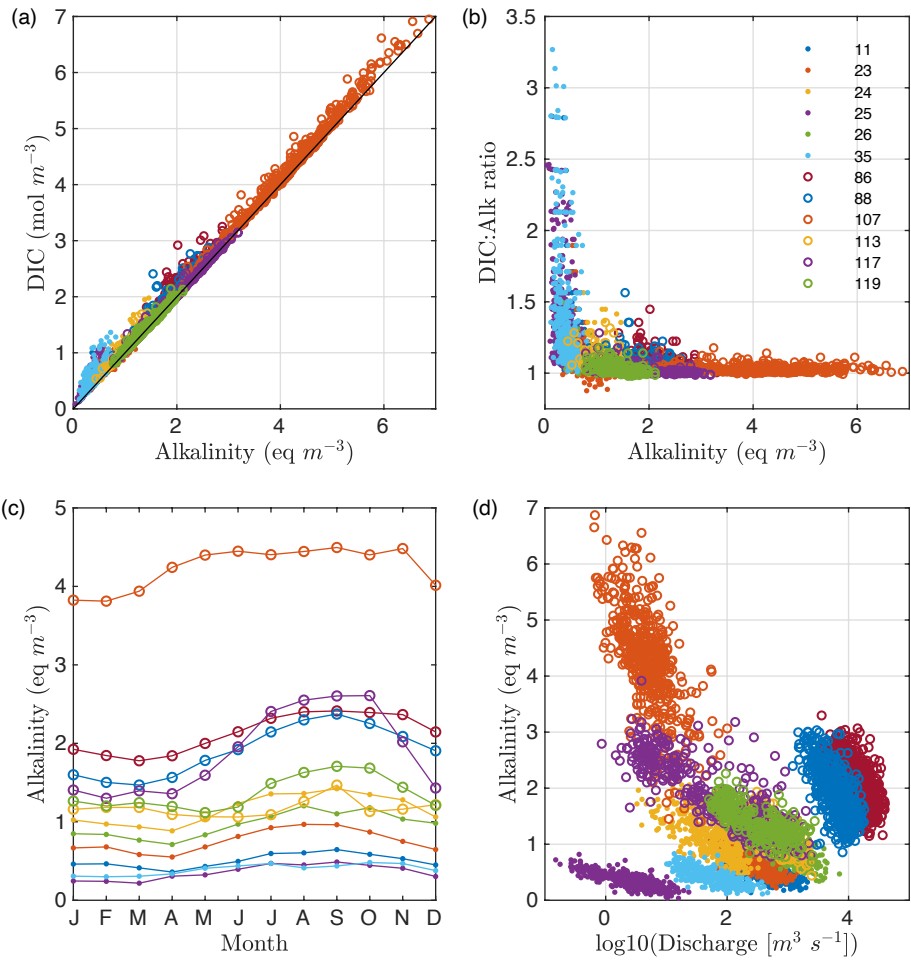

**Figure 7.** Carbon system patterns for 12 selected rivers: (a) DIC vs. alkalinity, and (b) DIC:Alk ratio vs. alkalinity; (c) monthly climatological patterns of alkalinity; and (d) alkalinity vs. logarithm of discharge. All patterns were derived for the 12 rivers with the largest number of records in the database: Connecticut (ID=11), Delaware (23), Schuylkill (24), Choptank (25), Susquehanna (26), Neuse (35), Mississippi (86), Atchafalaya (88), Santa Ana (107), Sacramento (113), Eel (117), and Klamath (119).

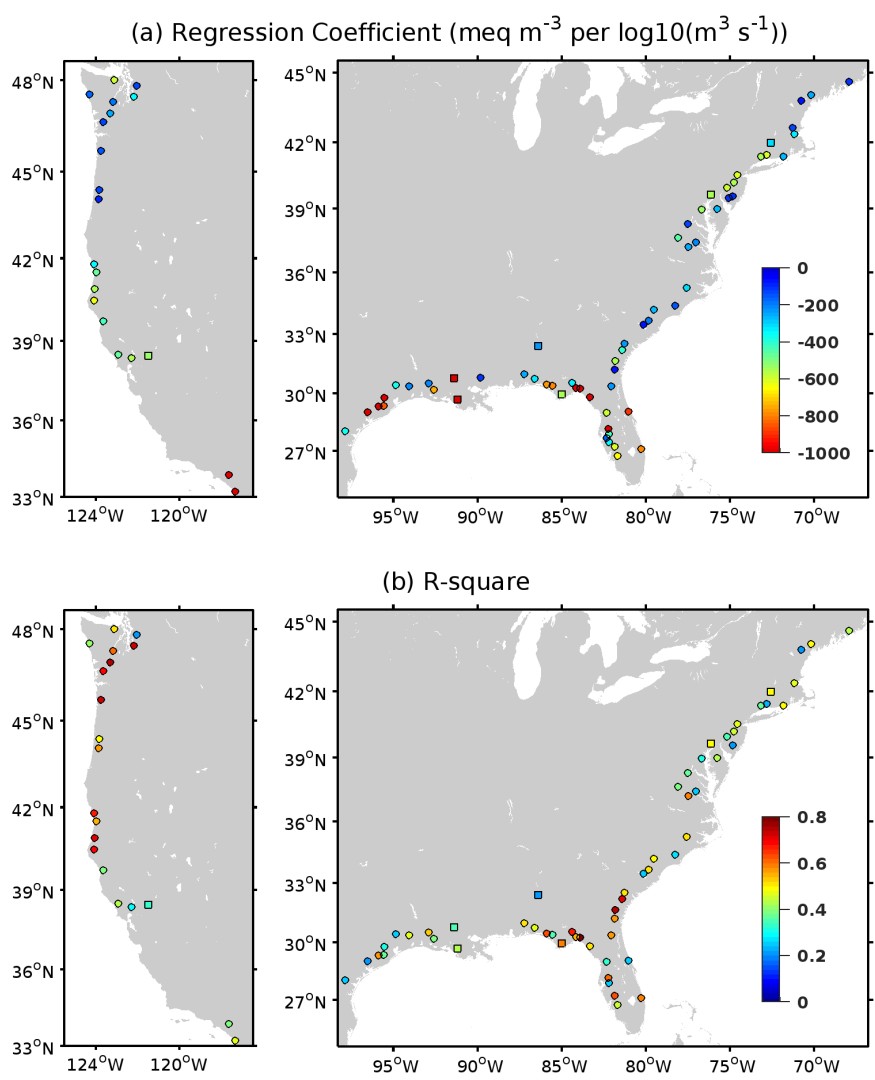

**Figure 8.** (a) Regression coefficient and (b) coefficient of determination ($R^2$) for the adjusted linear regressions between alkalinity and the logarithm of discharge (colored dots and squares). Patterns are shown only for the stations that have a significant regression coefficient, an $R^2$ greater than 0.2, and more than 30 observations (76 out of 140 stations). Squares (dots) represent river stations with a mean discharge greater (smaller) than 500 $m^3$ $s^{-1}$.



## 4 Data availability

The river chemistry data product is available in netCDF format at NOAA/NCEI with a DOI of
https://doi.org/10.25921/9jfw-ph50 and NCEI accession number 0260455 (Gomez et al., 2022). For each of the selected river
stations, we provide monthly time series and climatologies for each variable. Excel spreadsheets reporting the USGS
parameters used to generate the dataset, the station and river mouth locations (latitude/longitude), the number of records in the
series, and the first and last year in the series, are also provided in the dataset.

## 5 Summary and conclusion

Retrieving data from the USGS Water Quality database has complexities, such as identifying nearshore sites for coastal studies
(USGS contains more than 2,400 sites across the United States, many in inland waters that are not directly relevant to coastal
ocean analyses), or integrating water quality parameters to characterize biogeochemical properties (water properties are usually
described by more than one USGS parameter). Thus, a user not familiar with the USGS database may require considerable
time and effort identifying river sites and parameters. We facilitate this task, providing an integrated river chemistry and
discharge dataset for 140 USGS nearshore sites, which contains relevant variables to characterize biogeochemical and water
fluxes (land-to-ocean) along the U.S. West, East and Gulf of Mexico coasts. RC4USCoast includes data for alkalinity, pH,
nutrients, and novel estimates of river DIC. River mouth location (longitude, latitude) is reported for each USGS sites, which
expedites the data integration in coastal biogeochemical studies. The main goal is to fill a gap for river carbonate chemistry
products, as necessary inputs for regional model simulations that include ocean biogeochemistry. We also note the utility of
this product for skill assessment of hydrologic and riverine chemistry models estimating discharge and nutrient loading patterns
resulting from climate and land use activities (e.g., Lee et al., 2019). Patterns in RC4USCoast show distinct regional features
for alkalinity and DIC. The average and standard deviation of the DIC:Alk ratio increased in low alkalinity rivers, and both
alkalinity and DIC concentration were inversely related to river discharge. Our results revealed a significant spatiotemporal
variability in carbon chemistry, which can play a significant role on coastal biogeochemical dynamics.

**Author contributions:** FAG retrieved the USGS dataset and generated the monthly series and climatological data for the
selected sites and variables. AR produced the files in netCDF format. All authors contributed to the writing of the paper.

**Competing interests:** The authors declare that they have no conflict of interest.

**Acknowledgements:** This article was supported by NOAA's Ocean Acidification Program, Climate Program Office, and
Atlantic Oceanographic and Meteorological Laboratory.

**Financial support:** The research was conducted under NOAA's awards to the Northern Gulf Institute (NA16OAR4320199).





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
