# Peer review of "RC4USCoast: A river chemistry dataset for regional ocean model applications in the U.S. East, Gulf of Mexico, and West Coasts"

_Earth System Science Data, 2022_

## Author Response (AR1)

We highly appreciate the reviewers for their valuable comments and suggestions. Below, we have included our response (in blue color).

**Response to reviewer 1**

River inputs are critical for studies of the coastal ecosystem. Specifically, the regional modeling community needs standardized data input for both streamflow and water quality. We used Mississippi as an example and examined the data to find it is generally consistent with the river inputs we have composed based on USGS measurements regarding river discharge, nutrient, and carbonate variables. My only suggestion is that currently, the data stop in 2020, and I would encourage the authors to continuously extend the database in an up-to-date mode.

Following the reviewer suggestion, we have updated the database until 2022. We will also continue updating the NCEI archive database in the future on an annual basis.

Tables, Figures, and calculations were updated as we added two years to the dataset.

**Response to reviewer 2**

The RC4USCoast package is a useful compilation of river carbonate chemistry and discharge data for coastal ocean modeling purposes. The manuscript is clearly written and the data quality is excellent based on the best available USGS results.

A few minor comments:

P3 L63-64, the expression is confusing, you really meant that using TA and pH to estimate DIC may overestimate the latter because of the inclusion of non-carbonate alkalinity. Actually, non-carbonate alkalinity itself is not precise, borate isn't a part of carbonate alkalinity as well.

We agree that the expression was confusing. We also recognize that it is more appropriate to use the term organic alkalinity instead of non-carbonate alkalinity. Therefore, we have made a few changes to make the point clearer (lines 58-67):

Following Stets and Striegl (2012), we assumed that (*i*) particulate inorganic carbon is small; thus, filtered and unfiltered measurements of alkalinity are nearly the same, and (*ii*) organic alkalinity represents a small fraction of total alkalinity. A comparison between filtered and unfiltered measurements of alkalinity does not show significant differences (Fig. 2a); thus, biases associated with the first assumption are negligible. The second assumption is required because including the organic alkalinity fraction in the total alkalinity term used to derive DIC lead to some DIC overestimation. This could be a problem in low alkalinity rivers with high concentration of organic matter, as the latter contains anionic functional groups that can contribute to alkalinity (Hunt et al., 2011). Stets and Striegl (2012) discussed this issue further and showed that organic alkalinity usually represents less than 10% of total alkalinity in U.S. rivers, not producing important biases in the DIC calculations. Consistently, a comparison between measured DIC and the calculated DIC reveals a good agreement, with no evident bias in the residuals of the least square model (Fig. 2b).

Throughout the text, please change carbon chemistry to carbonate chemistry, please be precise.

We changed the term as suggested.

Table 1, the variable DIC should have both measured (a small fraction) and calculated, need to be complete.

We did not include the observed DIC in the original RC4USCoast version. But following the reviewer observation, we have include it in the updated database (and add it to Table 1).

P6 L116, DIC:Alk = 3.5 as cutoff for throwing out outliers, please explain.

Given the uncertainty associated with the DIC calculation, with a potential of DIC overestimation in low alkalinity rivers, we set an upper limit to the DIC:Alk ratio, based on ranges reported by Moore-Maley et al. (2018). The calculated DIC records greater than 3.5 times alkalinity were then removed from the database. The removed records correspond to a very minor fraction of the total monthly records (3.7%), mainly associated with the rivers, Toms, Satilla, St. Marys, and Blackwater.

This aspects are mentioned in the new manuscript version (lines 122-125):

Additionally, an upper threshold of 3.5 was used for the DIC to alkalinity ratio (DIC:Alk ratio), based on values reported by Moore-Maley et al. (2018). Calculated DIC records linked to DIC:Alk ratios greater than 3.5 were then removed. This was a very minor fraction of the total monthly DIC records (3.7%), mainly associated with low alkalinity's values in the Toms, Satilla, St. Marys, and Blackwater rivers.

P9 L144, "Blackwater". it seems that there are more than one Blackwater Rivers in the US, please specify its location. In viewing the data, this is a very small tributary in Alabama to another small river, not into the coast directly. Please reexamine the listed rivers and make sure they actually contribute to the coastal ocean.

We are referring to the Blackwater River in the Florida Panhandle, which discharges into Blackwater Bay (close to Pensacola Bay). We have updated Supplementary Table S1, providing a more accurate location for the Blackwater and Yellow mouth locations.

There are eight different parameters that can be interpreted as alkalinity more or less. For cases that these results do not agree (which occurs actually quite often), please explain how you merged the data.

Following a similar approach used in previous studies (e.g., Raymond et al, 2008; Stets and Striegl, 2012; Kaushal et al., 2013; Stets et al., 2014), we merged different USGS parameters for alkalinity to derive the monthly alkalinity series and climatologies. Although some discrepancies between parameters could be found, overall, the differences were minor. This can be seen in a comparison between the monthly alkalinity derived from parameter 00440 ($A_{00440}$: Acid neutralizing capacity, unfiltered, field), which was the most abundant in the database, and the other alkalinity parameters, which we have included in the Supplement. We also included a comparison for the USGS parameters linked to $NO_3$, $NH_4$, and $PO_4$.

In section 2.4 (lines 114-116) we have indicated:

Using more than one parameter for alkalinity, $NO_3$, $NH_4$ and $PO_4$ was a reasonable option to improve the spatiotemporal representation of the patterns, as concentration differences between parameters, during overlapping periods, were minor and did not reveal evident biases (Figs. S2 and S3 in the Supplement).

[Figure]

**Figure S2.** Scatterplots showing the association between the alkalinity parameter 00410 ($A_{00410}$; x-axis) and other alkalinity parameters (y-axis) in the USGS database: (a) $A_{00419}$, (b) $A_{29801}$, (c) $A_{39036}$, (d) $A_{39086}$, (e) $A_{90410}$, (f) $A_{00440}$, (g) $A_{00453}$).

[Figure]

**Figure S3.** Scatterplots showing the association between (a) nitrate parameters 00618 and 71851, (b) ammonia parameters 00608 and 71846, and (c) phosphate parameter 00660 and 00671.

**References**

Hunt, C. W., Salisbury, J. E., and Vandemark, D.: Contribution of non-carbonate anions to total alkalinity and overestimation of $p$CO$_2$ in New England and New Brunswick rivers, Biogeosciences, 8, 3069–3076, https://doi.org/10.5194/bg-8-3069-2011, 2011.

Kaushal, S.S., Likens, G.E., Utz, R.M., Pace, M.L., Grese, M., and Yepsen, M.: Increased river alkalinization in the Eastern US. *Environmental science & technology*, *47*(18), 10302-10311, https://doi.org/10.1021/es401046s, 2013.

Moore-Maley, B. L., Ianson, D., and Allen, S. E.: The sensitivity of estuarine aragonite saturation state and pH to the carbonate chemistry of a freshet-dominated river, Biogeosciences, 15, 3743–3760. https://doi.org/10.5194/bg-15-3743-2018, 2018.

Raymond, P., Oh, NH., Turner, R. *et al.*: Anthropogenically enhanced fluxes of water and carbon from the Mississippi River. *Nature* **451**, 449–452 (2008). https://doi.org/10.1038/nature06505

Stets, E. G., and Striegl, R. G.: Carbon export by rivers draining the conterminous United States, Inl. Waters, 2, 177–184, https://doi.org/10.5268/IW-2.4.510, 2012.

Stets, E.G., Kelly, V.J. and Crawford, C.G.: Long-term trends in alkalinity in large rivers of the conterminous US in relation to acidification, agriculture, and hydrologic modification. *Science of the Total Environment*, *488*, pp.280-289, https://doi.org/10.1016/j.scitotenv.2014.04.054, 2014.